Application of ultrasound guidance in the oral and maxillofacial nerve block

Cao Zhiwei
Zhang Kun
Hu Liru
Pan Jian jianpancn@scu.edu.cn
Department of Oral Surgery, State Key Laboratory of Oral Diseases & National Clinical Research Center for Oral Diseases & West China Hospital of Stomatology, Sichuan University , Chengdu , Sichuan , China
Husein Adam
Electronic publication date: 2021 Nov 26
Publication date: 2021
Volume: 9
Electronic Location ID: e12543
Received 2021 Jul 7; Accepted 2021 Nov 4
Copyright: ©2021 Cao et al.
Copyright year: 2021
Copyright holder: Cao et al.
License: This is an open access article distributed under the terms of the Creative Commons Attribution License, which permits unrestricted use, distribution, reproduction and adaptation in any medium and for any purpose provided that it is properly attributed. For attribution, the original author(s), title, publication source (PeerJ) and either DOI or URL of the article must be cited.
License URL: https://creativecommons.org/licenses/by/4.0/

Keywords: Nerve block, Ultrasound, Mandibular nerve, Maxillary nerve

Funding: Sichuan Academic and Technical Leader Training Foundation (2017-A) Sichuan Cadres Health Care Project (2019-901) Research and Develop Program, West China Hospital of Stomatology Sichuan University LCYJ2019-1 This study was supported by Sichuan Academic and Technical Leader Training Foundation (2017-A); Sichuan Cadres Health Care Project (2019-901). Research and Develop Program, West China Hospital of Stomatology Sichuan University, (LCYJ2019-1). The funders had no role in study design, data collection and analysis, decision to publish, or preparation of the manuscript.

==============================
Introduction

Nerve block technology is widely used in clinical practice for pain management. Conventional nerve localization methods, which only rely on palpation to identify anatomical landmarks, require experienced surgeons and can be risky. Visualization technologies like ultrasound guidance can help prevent complications by helping surgeons locate anatomical structures in the surgical area and by guiding the operation using different kinds of images. There are several important and complex anatomical structures in the oral and maxillofacial regions. The current article reviews the application of ultrasound guidance in oral and maxillofacial nerve blocks.

Methods

We searched the literature on the use of ultrasound guidance for the main nerve block techniques in the oral and maxillofacial regions using both PubMed and MEDLINE and summarized the findings.

Results and Discussion

A review of the literature showed that ultrasound guidance improves the safety and effectiveness of several kinds of puncture procedures, including nerve blocks. There are two approaches to blocking the mandibular nerve: intraoral and extraoral. This review found that the role of ultrasound guidance is more important in the extraoral approach. There are also two approaches to the blocking of the maxillary nerve and the trigeminal ganglion under ultrasound guidance: the superazygomatic approach and the infrazygomatic approach. The infrazygomatic approach can be further divided into the anterior approach and the posterior approach. It is generally believed that the anterior approach is safer and more effective. This review found that the effectiveness and safety of most oral and maxillofacial nerve block operations can be improved through the use of ultrasound guidance.

Introduction

The precise goal of a nerve block can be described as “put the right dose of the right drug in the right place” (Denny & Harrop-Griffiths, 2005). Conventional nerve localization methods are based on the identification of anatomical landmarks via palpation on the surface of the body. This technique relies heavily on the experience of the surgeons, and such localization methods are not suitable for patients with anatomical variations in nerves and blood vessels or patients with no perceived anatomical landmarks on the surface of the body. With the development of visualization technologies like ultrasound, CT, and MRI, surgeons can now observe the distribution and direction of nerves and blood vessels. Visualization technologies can greatly reduce the risk of adverse reactions and improve the efficiency of puncture procedures.

Ultrasound guidance is widely used in nerve block procedures for upper and lower limbs and spine and celiac nerves. Ultrasound also has a wide range of applications in the oral and maxillofacial region, including ultrasonic scaling, ultrasound diagnosis, ultrasound-guided puncture procedures, and ultrasound thermo-chemotherapy (Fig. 1), but there are relatively few studies on ultrasound-guided oral and maxillofacial nerve blocks. However, nerve block procedures in oral and maxillofacial region also have certain risks. For example, it is reported that routine inferior alveolar nerve block will cause damage to the tongue nerve and the inferior alveolar nerve, the incidence is between 1:26,762 and 1:160,571 (Pogrel & Thamby, 2000), and intravascular needle entrance with aspiration positive can be found in 15.3% of inferior alveolar nerve blocks which can lead to vascular damage and hemorrhage and sometimes intravascular injection (Blanton & Jeske, 2003; Taghavi Zenouz et al., 2008). In addition, oral and maxillofacial nerve block can also cause facial nerve palsy, transient amaurosis, temporary blindness, sudden unilateral deafness, total body hemiparesis, due to incorrect puncture and local anesthetic injection (Crean & Powis, 1999). The current article presents a review of the progress in the application of ultrasound guidance in oral and maxillofacial nerve blocks. This article aims to increase physician awareness of visualization techniques when performing nerve blocks in the maxillofacial region in order to reduce injuries and improve the efficiency of nerve blocks.

Figure 1 Application of ultrasound in the oral and maxillofacial region.

(A) Ultrasound Diagnostic System. (B) Ultrasonic scaler tip. (C) Ultrasound used to scan and diagnose the examination site. (D) Ultrasound guided puncture operation. (E) Ultrasound thermo-chemotherapy. (F) Ultrasonic scaling. (G) Ultrasonic root canal irrigation.

This article is intended for oral and maxillofacial surgeons, dental emergency doctors, and neurosurgeons, providing them with more visualization technology choices when performing nerve blocks or neuralgia treatment in daily clinical practice. It is our hope that these providers can develop more safe and effective treatment approaches with the help of visualization technology.

Methods

Literature search strategy

The present study was conducted following the Preferred Reporting Items for Systematic Reviews and Meta-Analyses (PRISMA) statement (Moher et al., 2009). This study comprehensively searched the PubMed and MEDLINE databases, using the following keywords or Medical Subject Heading (MeSH) terms. “OR” and “AND” operations are used to search the literature on the different nerves and guidance methods. For instance, when searching publications on ultrasound guidance used in inferior nerve blocks, the search terms used were “ultrasound guidance” OR “US” or “ultrasound-guided” OR “US-guided” AND “inferior alveolar nerve” OR “IANB”. Since studies involved both the inferior alveolar and the mandibular nerve, “mandibular nerve” was searched at the same time. Since no reviews similar to the current one is found, this study did not impose restrictions on the year of publication.

Study selection criteria

The studies were independently reviewed by two authors and any discrepancies were resolved by discussion and consensus. Studies were included based on the following criteria: (1) written in the English language; (2) had human subjects (including autopsy studies); and (3) were a clinical trial, cohort study, narrative study, or case report. Studies that were excluded were: (1) not written in English; (2) did not have human subjects; and (3) did not use visualization technology but used other neural localization technology to guide nerve block procedures. In total, 34 relevant publications were included in review including 3 existing review articles (Fig. 2).

Figure 2 Study selection process.

Assessment of levels of evidence

The evidence levels of the articles were assessed by the first author using the 2014 version of the Joanna Briggs Institute (JBI) Levels of Evidence. Evidence was ranked into one of five levels: level 1 = very high, level 2 = high, level 3 = moderate, level 4 = low and level 5 = very low.

Results

Thirty-four included studies are grouped according to the target nerves in this study. It was found that the most common target nerves for nerve block procedures in the oral and maxillofacial region are the inferior alveolar nerve, the mandibular nerve, the maxillary nerve and the trigeminal ganglion, and a nerve block of the mandibular nerve and the inferior alveolar nerve are frequently done at the same time, so these two groups in this article were merged. For blocks of the same nerve, different approaches are grouped. In addition to the target nerves mentioned above, there were also studies describing ultrasound-guided nerve blocks of other nerves including the infraorbital nerve and the supraorbital nerve. These were included in the ”other nerves” section in this article.

The main anatomical structures and nerves involved in the current article are shown in the lateral view of the skull in Fig. 3.

Figure 3 Anatomy structure of part of the skull in lateral view.

(A) Lateral view of part of the skull with mouth closed; (B) Lateral view of part of the skull after the zygomatic arch is removed with mouth closed; the pterygopalatine fossa is covered with coracoid process; (C) Lateral view of part of the skull where the zygomatic arch and mandible are transparent; (D) Lateral view of part of the skull after the zygomatic arch is removed with mouth opened; the pterygopalatine fossa can be seen. (E) Enlarged image of the pterygopalatine fossa.SB: sphenoid bone; TB, temporal bone; CB, Cheekbone; ZA, zygomatic arch; TG, Trigeminal ganglion; FR, foramen rotundum; FO, foramen ovale; CP, coracoid process, LPP, lateral pterygoid plate; PF, pterygopalatine fossa; MdN, mandibular nerve; MxN, maxillary nerve; LN, lingual nerve; IAN, inferior alveolar nerve; PSAN, posterior superior alveolar nerve; PG: pterygopalatine ganglion.

The mandibular nerve and the inferior alveolar nerve

Intraoral injection is the most commonly used method for inferior alveolar nerve blocks (IANB) in clinical practice. The puncture location is identified using several surface landmarks like the pterygomandibular ligament and the buccal fat pad. When practicing this approach, the needle is placed 1 cm above and parallel to the mandibular occlusal plane. Surgeons administer anesthesia to the inner surface of the mandibular ramus, and then the inferior alveolar nerve can be successfully blocked through the diffusion of anesthetic drugs (Aggarwal, Singla & Kabi, 2010). This method is widely used clinically, but studies have shown that its success rate is affected by the specific occasion, which means different types of dental treatment have different requirements for IANB (Abdallah, Macfarlane & Brull, 2016). Studies have proposed different measures to improve the success rate, such as supplementary infiltration anesthesia, changing the anesthetic drug dose (Milani et al., 2018), or using ultrasound-guidance methods. Ultrasound-guided IANB can be divided into intraoral and extraoral approaches.

Intraoral approach

In the intraoral approach, both the ultrasound probe and the injection needle are placed in the mouth. Hannan et al. (1999) compared intraoral ultrasound-guided IANB to conventional techniques. When comparing the degree of anesthesia of the dental pulp in one side of the mandible, it was found that although the needle tip can be accurately placed around the nerve using ultrasound guidance, no significant difference was found in the success rate between the two techniques. The author also pointed out that the indicator of a successful block used in the study was “pulpal anesthesia”, which is different from the commonly used “lip and tongue numbness”. Also, in this study, the inferior alveolar nerve could not be located using ultrasound, so the position of the inferior alveolar artery was used to locate the nerve (Hannan et al., 1999). Chanpong et al. (2013) were able to identify the inferior alveolar nerve using a new type of hockey stick-shaped ultrasound probe placed at the patient’s pterygomandibular ligament. The average scanning time required to locate the left inferior alveolar nerve was only 19.6 s and the scanning time required to locate the right inferior alveolar nerve was 30.5 s. The subjects stated that the probe did not cause any significant discomfort. In addition, the author successfully performed IANB on cadavers using this method by injecting a dye in order to simulate the diffusion of anesthesia drugs around the nerve when performing this procedure (Chanpong et al., 2013).

In summary, there are several criteria to evaluate the success of traditional IANB, and accurately placing the needle around the nerve is not the only factor that affects anesthesia. Future studies addressing the safety of this method are needed as there are only a few studies (including basic research and clinical research) on the intraoral approach to ultrasound-guided inferior alveolar nerve blocks. It is too early to conclude that the introduction of ultrasound has any effect on the success rate of IANB in the intraoral approach. More studies are needed to explore whether ultrasound can guide this operation.

Extraoral approach

In the extraoral approach of ultrasound-guided inferior alveolar nerve (or mandibular nerve) blocks, the ultrasound probe and the puncture point are located outside of the mouth. The most common purpose behind the extraoral approach is to inject the anesthetic drugs into the pterygomandibular space. The relative positions of the ultrasound probe and the needle tip, and the correct recognition of the anatomical structures from the ultrasound images are important for this procedure. Kumita et al. reported a type of extraoral approach to performing an ultrasound-guided maxillary and inferior alveolar nerve block to induce analgesia after orthognathic surgery. In this method, the ultrasound probe was placed caudad to the zygomatic arch to observe the maxillary artery in the pterygomandibular space. The selected injection site was just around the maxillary artery to help the anesthetic drug infiltrate the inferior alveolar nerve. This study also used the position of the maxillary artery to locate the nerve in the ultrasound images (Kumita, Murouchi & Arakawa, 2017). Using the same method, Kojima et al. (2020) performed ultrasound-guided extraoral approach IANB in patients having drug-related osteonecrosis and undergoing mandibular resection under general anesthesia. The after-surgery analgesic effect was better in the experimental group compared to the control group of patients who did not receive IANB, and the total amount of opioids used in the experimental group was significantly less than the amount used by patients in the control group (Kojima et al., 2020). The above studies showed that ultrasound-guided injection of local anesthetics into the pterygomandibular space can achieve satisfactory results in inferior alveolar nerve (and mandibular nerve) blocks. As an alternative to injecting the anesthetic drugs into the pterygomandibular space, Kampitak, Tansatit & Shibata (2018b) performed a new ultrasound-guided selective mandibular nerve block in cadavers, called the lateral pterygoid plate approach, where the drugs were injected into the base of the skull. During the surgery, the cadaver’s mouth was wide open, the posterior and superior edges of the lateral pterygoid plate were identified using ultrasound, and the adjacent mandibular nerve and its branches were successfully stained by injecting a dye, indicating that a successful nerve block can be performed in the same way (Kampitak, Tansatit & Shibata, 2018b). Tsuchiya et al. (2019) also used a similar approach, in which the lateral pterygoid plate was used as a landmark. In ten cases of parotidectomy, low-molecular weight dextran and local anesthetics were used to block the mandibular nerve under ultrasound guidance, successfully reducing the involuntary movement of the muscles due to surgical stimulation during operation, thus reducing the need for general anesthesia (Tsuchiya et al., 2019).

In summary, there are relatively more studies on the extraoral approach of ultrasound-guided alveolar nerve (or mandibular nerve) blocks compared to intraoral approach. In some variations of the extraoral approach of the mandibular nerve or IANB, patients do not need to open their mouths. Since there are some patients who require a mandibular nerve or IANB that cannot open their mouths due to trauma or pain, the extraoral approach of the alveolar nerve (or mandibular nerve) block is sometimes necessary. However, in the extraoral approach, because the needle goes deeper and the adjacent structures are more complex, the ultrasound can better guide the needle. Jain et al. performed closed-mouth high-position mandibular nerve blocks known as the Vazirani-Akinosi method (Prabhu Nakkeeran et al., 2019) and extraoral ultrasound-guided mandibular nerve blocks in patients with pain and trismus due to fracture or acute pain, before administering general anesthesia. To check for differences between these two block methods, the visual analogue scale (VAS) pain scores and the degree of relief from trismus before and after the block for each patient was compared. The results suggest that ultrasound can help accurately locate nerves and blood vessels anterior to the condyle and assist with injecting anesthetics into the correct location. The ultrasound-guided group of patients required fewer anesthetic drugs, had fewer adverse reactions, and showed better anesthetic effects and higher success rates of block (Jain et al., 2016). The comparison of the different extraoral approaches of IANB is shown in Table 1 and Fig. 4.

Table 1 Comparison of different approaches of ultrasound guided mandibular nerve and inferior alveolar nerve blocks.

Approach	Authors	Subjects	Levels of evidence	Position of ultrasound probe	Information of the probe	Puncture point
/Inject point	Ultrasound- guided puncture method	Main advantages	Possible disadvantages	
Intraoral	Hannna, 1999	Patients	1	The oral mucosa on medial aspect of the mandibular ramus	Acuson EV519
frequency unclear	Intraoral/
IAN at PS	In plane	Locate the nerve accurately,
Avoid inadvertent intraneural injection and entry into surrounding structures	The success rate was not significantly improved; Possible increase in operating time	
	Chanpong, 2013	Patients and cadavers	4	Pterygomandibular raphe	HST15-8/20 linear probe
8- to 15-MHz	Intraoral/ IAN at PS	–			
Extraoral	Kumita, 2017; Kojima, 2020	Patients	4	Caudad to ZA.	linear probe
5–12 MHz	Below the ZA/ Maxillary artery	Out of plane		Large degree of mouth opening	
	Kampitak, 2018	Cadavers	5	Transversely below the
ZA.	SonoSite X–Porte 8–3 MHz	Below the ZA/ The Posterior border of the LPP	In plane		Large degree of mouth opening	
	Tsuchiya, 2019	Patients	4	Below and parallel to the ZA	curvilinear probe
4.5 MHz	- /Dorsal edge of LPP	–			
	Jain, 2016	Patients	1	Superior to the mandible (linear ultrasound probe); below the zygoma and anterior to the mandibular condyle (cardiac probe)	12 L-RS linear probe
8–13 MHz
Or
cardiac probe
2.8–4 MHz	Superior (linear ultrasound probe) or posterior (cardiac probe) to the probe/ Mandibular nerve	Out of plane (linear ultrasound probe)
In-line (cardiac probe)	No need to open mouth
Reduce the dosage of local anesthetics and avoid vascular puncture		
Notes.

ZA, zygomatic arch; LPP, lateral pterygoid plate; PS, pterygomandibular space.

Figure 4 The needle position of different methods of performing an ultrasound-guided maxillary nerve block in extraoral approach with mouth opened.

1: The trajectory of the needle is perpendicular to the sagittal plane to reach the PMS, described by (Kumita, Murouchi & Arakawa, 2017) 2: The needle reaches the posterior edge of the LPP in the infra-zygomatic approach described by (Kampitak, Tansatit & Shibata, 2018a)

Maxillary nerve

The extraoral approach is the most common approach used in ultrasound-guided maxillary nerve blocks. In this procedure, the ultrasound probe is usually placed below the zygomatic arch. The extraoral approach can be further divided into the infra-zygomatic and supra-zygomatic approaches depending on the location of the puncture point (Anugerah, Nguyen & Nader, 2020).

Supra-zygomatic approach

Sola et al. (2012) and Chiono et al. (2014) performed bilateral maxillary nerve blocks using ultrasound guidance and the supra-zygomatic approach for pain management in infants who underwent cleft palate repair surgery. In ultrasound images, anatomical structures in the pterygopalatine fossa and needle tip positions could be clearly distinguished in real-time images. The anesthetic drug was injected into the pterygopalatine fossa using ultrasound guidance, and the surgeons could control the spread of the anesthetic over time. The arterial pulsation could also be monitored to help avoid the risk of vascular puncture. Although the maxillary nerve could not be directly identified using ultrasound images, the success rate and safety of the maxillary nerve block in this study were both improved. After the surgery, all patients with a maxillary nerve block had better pain management and used fewer analgesic drugs (Chiono et al., 2014; Sola et al., 2012). Another study performed maxillary nerve blocks during maxillary osteotomy operations in adult patients using supra-zygomatic approach and determined that this method was safer as it avoids the risk of penetrating the orbital contents through the infraorbital fissure (Bouzinac et al., 2014). Echaniz et al. used this method to conduct a study on cadavers to identify the anesthetic drug dose required for a maxillary nerve block. The results showed that only 1 mL of liquid was needed to cover the nerve surface (Echaniz et al., 2019). In the study by Kumita, Murouchi & Arakawa (2017) a maxillary nerve block was performed right after a mandibular nerve block, and the pterygoid fossa could be observed by adjusting the position of the probe. In this study, the insertion point was located at the angle formed by the superior edge of the zygomatic arch and the posterior orbital rim, and the injection site was at the lateral pterygoid plate. The authors pointed out that using ultrasound-guided maxillary and inferior alveolar nerve blocks at the same time significantly improved the effectiveness of perioperative analgesia during gnathoplasty (Kumita, Murouchi & Arakawa, 2017).

Infra-zygomatic approach

The supra-zygomatic approach mentioned above is considered a safer approach by many scholars, but normally it can only be performed using out-of-plane techniques, which are difficult. Moreover, due to the occlusion of the zygomatic arch, there is a period when the needle is seemingly invisible in the ultrasound and the surgeons have to rely on their experience (Anugerah, Nguyen & Nader, 2020). So, there are infra-zygomatic approach which can solve these problems.

(Kampitak, Tansatit & Shibata, 2018a) introduced a kind of ultrasound-guided maxillary nerve block using the infra-zygomatic approach in cadavers, using ultrasound images of the posterior edge of the maxilla and the lateral pterygoid plate as landmarks. In this study, while the mouth of the cadaver was wide open, the injection needle could approach the pterygopalatine fossa through the front edge of the lateral pterygoid plate. This method successfully simulated the block of the maxillary nerve, pterygopalatine ganglion, greater and lesser palatine nerves, and middle and posterior superior alveolar nerves by injecting a dye in place of an analgesic to see how the analgesic would spread in the nerves (Kampitak, Tansatit & Shibata, 2018a). The authors claimed that after administering general anesthesia, the degree of opening of the mouth could be increased (Kampitak & Shibata, 2019). In the infra-zygomatic approach, there are two ways for the needle to enter: the anterior and posterior approaches. Alfaro-de la Torre et al. (2019) compared these two approaches and found that the anterior approach, in which the needle goes front-to-back, can effectively avoid damage to important structures like the facial nerve, parotid gland, and maxillary artery. However, the anterior approach is more difficult to perform. In addition to the two approaches described above, (Takahashi & Suzuki, 2017) reported a novel infra-zygomatic approach, in which the needle passes from in front of the coronoid process. In this paper, this review refer to this method as “the coronoid approach”. The main advantages of this novel method are that the path of the needle is far away from the main blood vessels, and the needle would not advance to the pterygopalatine fossa but instead to the infratemporal crest, which is safer. The mandibular nerve can also be blocked at the same time by tilting the ultrasound probe slightly posteriorly and advancing the needle vertically. Patient discomfort could also be reduced since this method can be performed while the patient’s mouth is closed (Takahashi & Suzuki, 2017; Takahashi & Suzuki, 2018). Chang et al. (2017) and Ying & Du (2017) introduced another version of the infra-zygomatic approach where the needle goes between the coracoid process and the maxilla and easily passes through the fissura pterygomaxillaris to reach the pterygopalatine fossa, effectively avoiding the bone structures.

There are several studies on ultrasound-guided maxillary nerve blocks, but there is still no consensus on which method is better. The comparison of the different approaches to ultrasound-guided maxillary nerve blocking is shown in Table 2 and Fig. 5.

Table 2 Comparison of different approaches of ultrasound guided maxillary nerve blocks.

Approach	Authors	Subjects	Levels of evidence	Position of ultrasound probe	Puncture point and needle position/ Inject point	Informa-tion of the probe	Ultrasound- guided puncture method	Main advantages	Possible disadvantages	
Supra-zygomatic approach	Sola, 2012
Chiono, 2014
Bouzinac, 2014	Patients	3	Infra-zygomatic area, over the maxilla, with an inclination of 45° in both the frontal and horizontal planes	The angle formed by the superior edge of the ZA and the posterior edge of orbital rim./ The greater wing of the sphenoid at PF	linear array probe.
8–13MHz	Out of plane	Reduce postoperative analgesic dose
Help locate the needle and prevent the puncture of orbital contents	-	
	Echaniz, 2019	Cadavers	5							
Anterior approach (infra-zygomatic)	Kampitak, 2018	Cadavers	5	Transversely below the ZA and was tilted from the caudal to the cranial direction	-/ At the top of LPP at PF	curved array probe
3-8 MHz	In plane	By inserting the needle from front to back, avoiding puncture the parotid gland, the facial nerve and maxillary artery.	Large degree of mouth opening;
A very sharp needle entry angle is needed;
Sonographic visualization can be difficult	
Posterior approach
(infra-zygomatic)	Alfaro-de, 2019	Cadavers	5	–	-/-	Not clear	–	Easier to perform	It can possibly lead to injury to some vital organs, and needle pathway is longer	
Coracoid approach (infra-zygomatic)	Takahashi & Suzuki, 2017; Takahashi & Suzuki, 2018	Patients	4	Below the zygomatic process of the maxilla and tilted slightly in the superior direction.	Between the maxilla and coronoid process/ Infratemporal crest	Not clear	Out of plane	The needle does not enter into PF; Patients don’t have to open their mouth;
Mandibular nerve can be blocked at the same time	–	
Notes.

ZA, zygomatic arch; LPP, lateral pterygoid plate; PF, pterygopalatine fossa.

Figure 5 The needle position of different methods of performing an ultrasound-guided mandibular nerve block.

(A) The needle position of different methods of performing an ultrasound-guided maxillary nerve block while the mouth is closed or slightly opened. The dotted line indicates that the needle is blocked by the zygomatic arch or coracoid process. 1: the supra-zygomatic approach described by Sola et al. (2012); 2: the coracoid approach of the infra-zygomatic approach described by Takahashi & Suzuki (2017) (B) The needle position of different methods of performing an ultrasound-guided maxillary nerve block while the mouth is opened. 3: Posterior approach of the infra-zygomatic approach described by Nader et al. (2013a, 2013b) where the mouth is slightly opened; 4: Anterior approach of the infra-zygomatic approach described by Kampitak, Tansatit & Shibata (2018a) in which the anterior edge of LPP is used as the landmark for puncturing.

Trigeminal ganglion

An ultrasound-guided trigeminal ganglion puncture is similar to a maxillary nerve puncture. A nerve block of the trigeminal ganglion is used as a treatment for neuralgia of the trigeminal nerve and its branches (Allam et al., 2018). Nader et al. (2013a) injected anesthetic drugs and steroids under ultrasound guidance into the pterygopalatine fossa of fifteen patients who had trigeminal neuralgia and who had failed both surgery and drug treatments. The results showed that 80% of patients achieved complete analgesia in three branches of the nerve. The authors hypothesize that the high success of this procedure is because the pterygopalatine fossa communicates with the foramen rotundum and the supraorbital fissure (Nader et al., 2013a). The injection in the pterygopalatine fossa helps diffuse the anesthetic drugs to the middle cranial fossa through the foramen rotundum, thereby blocking the three branches of the trigeminal nerve (Nader et al., 2013a; Nader, Schittek & Kendall, 2013b). Although the trigeminal ganglion is situated relatively deeper, by adjusting the angle of the ultrasound probe and the injection needle, surgeons can still advance the needle through the upper head of the lateral pterygoid muscle and the pterygomaxillary fissure, and finally to the foramen rotundum through the pterygopalatine fossa (Chuang & Chen, 2015). A subsequent study used this ultrasound-guided puncture method on a patient with trigeminal neuralgia. The authors performed pulsed radiofrequency via the pterygopalatine fossa, achieving satisfactory analgesic effects, and no recurrence was observed (Nader et al., 2015). Kumar et al. (2018) used the same procedure on patients undergoing maxillofacial surgery for pain management, reducing the amount of opioids these patients used after surgery.

Ultrasound-guided trigeminal nerve block technology generally uses the infra-zygomatic approach. Zou et al. (2020) proposed a supra-zygomatic approach, where both the ultrasound probe and the puncture point are both located above the zygomatic arch. After identifying the landmark structures like the maxilla, the great wing of the sphenoid bone, and the pterygopalatine fossa using ultrasound imagery, the anesthetic drugs were injected onto the surface of the maxilla. The comparison of the skin sensation of the patients as well as MRI images both before and after the operation showed that this method helped spread the anesthetic drugs to the target nerves, and both the mandibular and maxillary nerves were blocked successfully. However, the blocking effect of the ophthalmic nerve was poor. This method is also more comfortable for patients as it does not require the mouth to be open. In-plane technology can also help reduce the difficulty of puncturing (Zou et al., 2020). The best way to perform an ultrasound-guided trigeminal nerve block, however, is still unclear. A comparison of the different approaches to ultrasound-guided maxillary nerve blocks is shown in Table 3 and Fig. 6.

Other nerves

Apart from the larger branches of the trigeminal nerve like the maxillary and the mandibular nerves, some small branches in the oral and maxillofacial region can also be blocked using ultrasound guidance (Allam et al., 2018). However, due to their superficial position, these nerves can be blocked easily using conventional landmark palpation, so the application of ultrasound has not been widely promoted in this area.

Michalek et al. (2013) used ultrasound to observe the position of the infraorbital foramen on a skull model. The authors concluded that it is feasible to block the infraorbital nerve under ultrasound guidance, and when the puncture point is located intraorally, the block can be performed using the in-plane technique because the path of the needle is longer (Michalek et al., 2013). Other studies used an ultrasound-guided infraorbital nerve block for patients with trigeminal neuralgia and isolated infraorbital neuralgia, and showed that rapid and satisfactory analgesia was achieved after surgery; although, neuralgia recurred in both studies (Takechi et al., 2015). The results from another randomized double-blind clinical trial of ultrasound-guided infraorbital nerve blocks suggest that dexmedetomidine combined with bupivacaine have a superior analgesic effect than dexamethasone combined with bupivacaine after cleft palate repair surgery (El-Emam & El Motlb, 2019).

Table 3 Comparison of different approaches of ultrasound guided trigeminal ganglion blocks.

Approach	Authors	Subjects	Levels of evidence	Position of ultrasound probe	Puncture point and needle position/Inject point	Informa-tion of the probe	Ultrasound- guided puncture method	Main advantages	Possible disadvantages	
Infra-zygomatic approach	Nader et al. (2013a)Nader et al. (2015); Chuang & Chen (2015)	Patients	4	Below the ZA, longitudinally on the side of face.
Below the ZA, longitudinally on the side of face and superior to mandibular notch, anterior to the mandibular condyle	Parallel to the probe/ Below the lateral pterygoid muscle, anterior to the LPP at PF	11-L transducer probe.
Frequency not clear	In plane
Out of plane	The subject’s mouth was just slightly opened	Mouth was kept open	
	Kumar, 2018	Patients	1			M-Turbo linear array probe.
7–12 MHz (				
Supra-zygomatic approach	Zou, 2020	Patients	4	Longitudinally on the side of the face just on the ZA and shifted cranially temporal fossa appeared and the ZA vanished in the ultrasound images.	Posterior side of the probe/ Maxilla surface	convex array probe
2- to 5-MHz	In plane	The needle does not enter into PF;
Patients don’t have to open their mouth.	Ophthalmic nerve cannot be blocked	
Notes.

ZA, zygomatic arch; LPP, lateral pterygoid plate; PF, pterygopalatine fossa.

Figure 6 The needle position of different methods of performing an ultrasound-guided Trigeminal Ganglion block.

Studies describe that an injection into the pterygopalatine fossa can make the anesthetic drugs diffuse through the foramen rotundum to the middle cranial fossa, thereby blocking three branches of the trigeminal nerve. 1: The supra-zygomatic approach described by Zou et al. (2020) 2:The infra-zygomatic approach described by Nader et al. (2013a, 2013b) Takahashi & Suzuki (2017)

Cadaver studies (Spinner & Kirschner, 2012) have demonstrated that ultrasound can also be used to guide injection into the supraorbital, infraorbital, and mental nerves. Luo, Lu & Ji (2018) and Ren, Shen & Luo (2020) studied the treatment of refractory supraorbital neuralgia with radiofrequency pulses and radiofrequency thermocoagulation, respectively, using ultrasound guidance. The results showed that both methods achieved satisfactory analgesic effects and no obvious adverse reactions were observed (Luo, Lu & Ji, 2018; Ren, Shen & Luo, 2020).

Hafeez et al. (2014) performed greater palatine nerve blocks with ultrasound guidance with a hockey stick-shaped ultrasound probe in patients and cadavers. They found that although ultrasound could locate the greater palatine artery and identify the greater palatine foramen and its direction effectively in the normal and edentulous maxilla, the block procedure could be challenging. The authors reported that with a more suitable size of ultrasound probe and a more suitable dose of local anesthetics, ultrasound may be used in pre-procedural localization to achieve a successful greater palatine nerve block (Hafeez et al., 2014).

In the maxillofacial region, although some nerves are superficial, they are accompanied by blood vessels, and the adjacent structures are complex. Ultrasound may play an important role in visualization and guidance of puncture operations. Ultrasound technology has the potential to significantly improve nerve blocks and the treatment of neuralgia.

Conclusion

The biggest advantage of ultrasound is its convenience and real-time imaging capabilities. These real-time images may help surgeons observe the blood vessels and nerves around an injection area and the diffusion of anesthetics in real-time, thereby improving the success rate and safety of puncture procedures. While there are relatively few studies on the application of ultrasound guidance in oral and maxillofacial nerve blocks and no widely recognized standards, this comprehensive literature review suggests the superiority of using ultrasound to guide nerve blocks in the oral and maxillofacial region.

Also, ultrasound technology is constantly developing and there have been several suggestions to enhance positioning abilities (such as needle visualization technology, endoscopic ultrasound, etc.) (O’Donnell & Loughnane, 2019). More research on ultrasound guidance is needed in the future. With the advancement of imaging technologies, more problems may be solved with support from clinical trials and medical practitioners.

Additional Information and Declarations

Competing Interests

Author Contributions

Data Availability

The authors declare there are no competing interests.

Zhiwei Cao conceived and designed the experiments, performed the experiments, analyzed the data, prepared figures and/or tables, authored or reviewed drafts of the paper, and approved the final draft.

Kun Zhang performed the experiments, authored or reviewed drafts of the paper, and approved the final draft.

Liru Hu analyzed the data, prepared figures and/or tables, and approved the final draft.

Jian Pan conceived and designed the experiments, authored or reviewed drafts of the paper, and approved the final draft.

The following information was supplied regarding data availability:

This is a review article.

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
