# Peer review of "Application of ultrasound guidance in the oral and maxillofacial nerve block"

_PeerJ, doi:10.7717/peerj.12543_

## Round 0.1 · original submission · Major Revisions

Dear Authors,

Please try to address all comments with corrections or rebuttals.

Please pay attention to the experimental design involving the search strategy, comments about the validity of the findings, evidence for the conclusions, and other important issues.

Reviewer 1 ·

Basic reporting

Well done to the authors for an interesting review. Overall it is well written and supplemented with comprehensive summary tables, clear figures and relevant references.

Proposed improvements:
1. The results and discussion in the abstract is too general. Authors may consider to specifically state the important findings from the literature review.
2. The result section can be improved with a descriptive summary of selected article types and parameters of interest (i.e nerve, approach, etc.) at the beginning of this section.

Experimental design

The design of this study is acceptable for a literature review.

For improvement:
1. In view of an almost similar literature search process according to PRISMA protocol, authors may add a flow chart (diagram) to show the process of literature selection.

Validity of the findings

The findings from the literature review are well summarized and reported. Excellent summary tables.

Additional comments

Additional comments:
1. Additional diagram on ultra sound application at the oral and maxillofacial region can be considered.
2. Brief discussion on study limitation is also valuable. This can be supplemented with a summary table of level of evidence for each selected article. Another alternative is to add column for 'Level of Evidence' next to 'Authors' column in all 3 tables.
3. Minor typographic error: 'In palne' in table 1 (first row).

Reviewer 2 ·

Basic reporting

Please correct the grammar. Commas are used injudiciously throughout the text.

Experimental design

The search strategy for the articles is not very clear. "Most of the papers were published after 2010" the statement is very subjective. Please rewrite it objectively.

Validity of the findings

Intra oral approach – inferior alveolar nerve. The authors have relied on insufficient evidence to conclude that there is limited utility in using ultrasound guided nerve block of inferior alveolar nerve. It is better to stress on the paucity of clinical studies while concluding the same.

Additional comments

There is insufficient information regarding the type of ultrasound probe and the frequency used in these studies

·

Basic reporting

The article is well written and in professional English.
The review have covered a wide range of the area and does not need any amendments or changes.

Experimental design

The article have reviewed the relevant papers and succeed to come up with a solid coherent article.

Validity of the findings

Findings and conclusions are well stated.

Additional comments

I highly recommend this article for publication.

---

## Round 0.2 · Minor Revisions

Thank you for the corrections that tremendously improved the manuscript. There are several minor corrections are still needed: 1) Please rephrase all the sentences with the words "we" and "they" (try to avoid in scientific writing); 2) A few places in the Introduction need references (please refer the file); 3) Please include the prevalence of injuries with the current techniques in the Introduction; 4) A citation at line 403 has no year (please include year); 5) Please follow the journal format of citation as per yellow highlight; 6) Minor spelling and grammar as highlighted in the manuscript. Thank you

Reviewer 2 ·

Basic reporting

The article meets the standards of the journal

Experimental design

no comment

Validity of the findings

no comment

Additional comments

The article can be published in its current form

---

## Round 0.3 · accepted · Accept

Dear authors,
Congratulations. Your manuscript is now accepted for publication. Best wishes